

# The potential biofertilization effect of $H_2$ is accompanied by a modest impact on the composition of microbial communities in the rhizosphere of common vetch

Diana Dip and Philippe Constant

Centre Armand-Frappier Santé Biotechnologie, Institut national de la recherche scientifique, Université du Québec, Laval, Québec, Canada

## ABSTRACT

**Background**. $N_2$-fixing nodules release molecular hydrogen ($H_2$) in the rhizosphere of legumes. The process activates $H_2$-oxidizing bacteria (HOB) in soil, leading to multiple effects on biogeochemical processes and a potential biofertilization effect. The legacy effect of the energy potential of $H_2$ on the soil microbial community structure and the population density of HOB has received little attention. The aim of the current study is to evaluate how the legacy effect of HOB, previously activated in soil microcosms exposed to elevated $H_2$ concentrations (e$H_2$), affects biomass production yield of common vetch (*Vicia sativa*), the abundance of HOB, and the composition of the rhizosphere microbiome.

**Methods**. Contrasting soil samples displaying more than 60% difference in $H_2$ oxidation activity were used as growth substrate for vetch. Soil microbial community composition and diversity were examined by bacterial 16S rRNA polymerase chain reaction (PCR) amplicon sequencing, and dry weight (DW) of the above- and below-ground biomass of vetch was analyzed to assess the impact of HOB enrichment on plant growth. The population density of high-affinity HOB was estimated by using the droplet digital polymerase chain reaction (ddPCR) technique to target the *hhyL* gene, encoding for the large subunit of group 1H/5 [NiFe]-hydrogenase.

**Results**. The abundance of HOB possessing group 1H/5 [NiFe]-hydrogenase was indistinguishable between the treatments, indicating that soil nutrient content (inorganic and organic carbon) and the energy potential of $H_2$ were insufficient to support their growth. *Aeromicrobium* spp. and *Ramlibacter* spp. were favored by e$H_2$ exposure at the activation stage, but their response was lost after vetch growth. The root biomass and the root/shoot ratio were reduced in soil conditioned with e$H_2$ compared to control soil exposed to ambient $H_2$, suggesting that the plant growth-promotion activity of HOB reduces root proliferation for nutrient prospection. These results provide new experimental evidence suggesting the biofertilization effect of $H_2$ is not universal and requires specific conditions that are yet to be identified.

Corresponding author
Philippe Constant,
philippe.constant@iaf.inrs.ca

## INTRODUCTION

$N_2$-fixing nodules of legume plants release $H_2$ as a by-product of the $N_2$ fixation reaction. In the case of symbiosis involving rhizobia without the [NiFe]-hydrogenase (*hup-* genotype), the release of $H_2$ creates a concentration gradient from approximately 10,000 ppmv at the soil-nodule interface to sub-atmospheric concentration (<0.5 ppmv) five cm away from the source (*Dong & Layzell, 2001*; *Piché-Choquette, Khdhiri & Constant, 2018*). At least 40 phyla of $H_2$-oxidizing bacteria (HOB) exploit the energy potential of $H_2$ (*Greening et al., 2016*). In carbon limitation conditions, HOB can survive by relying on $H_2$ as the sole energy source or by a mixotrophic strategy combining $H_2$ with organic carbon (*Bay et al., 2021*; *Greening et al., 2015*; *Islam et al., 2020*; *Liot & Constant, 2016*). According to metagenomic surveys, approximatively 12% of soil bacteria use $H_2$-derived electrons to fix $CO_2$ (*Xu et al., 2021*).

An elevated $H_2$ concentration in the rhizosphere of legume plants promotes the enrichment of HOB (*La Favre & Focht, 1983*). Incubating soil samples in modified atmospheres comprising different $H_2$ concentrations isolates the impact of $H_2$ on the composition and activity of soil microbial communities. Soil exposure to 250 nmol $H_2$ g(soil)$^{-1}$ h$^{-1}$ mimics $H_2$ fluxes from $N_2$-fixing nodules and enriches $CO_2$ fixation activity from HOB (*Stein et al., 2005*). The number of cells affiliated to Cytophaga-Flavobacterium-Bacteroides was also impacted by $H_2$, as shown by phylotype enumeration by fluorescence *in situ* hybridization analyses (*Stein et al., 2005*). Soil exposure to elevated $H_2$ concentrations promotes the activity of HOB degrading persistent organic pollutants in soil (*Xu et al., 2023*). The activation of HOB in soil exposed to $H_2$ for a few weeks led to variable impacts on the composition of soil microbial communities (*Khdhiri et al., 2017*; *Khdhiri et al., 2018*), calling into question the impact of $H_2$ on soil microbiota.

Soil fumigation by $H_2$ enhances plant growth. For instance, wheat biomass increased by 48% in soil samples exposed to a $H_2$-enriched atmosphere compared to control soil samples (*Dong et al., 2003*). This phenomenon is known as the $H_2$ fertilization effect (*Golding & Dong, 2010*). The exact mechanism is currently unknown, but some evidence suggests that it involves HOB activated by $H_2$, including *Variovorax* spp., displaying plant growth-promotion activity (*Maimaiti et al., 2007*). The current study was conducted to determine whether a high concentration of $H_2$ in soil activates and increases the population of HOB and if HOB activation modifies the composition of the soil microbial community and enhances the growth of *V. sativa*. These hypotheses were tested by setting up a two-stage experimental design. The first stage consisted of conditioning two contrasting HOB populations by exposing a sandy loam soil to either (i) sub-atmospheric (<0.5 ppmv) or (ii) supra-atmospheric (5,000 ppmv) $H_2$ concentrations. Soil exposure to $H_2$ lasted two months to favour the activation and enrichment of HOB in soils exposed to high $H_2$ concentration. The second stage consisted of growing *V. sativa* using the enriched soils as a substrate,

without further exposure to the modified atmosphere. The plant biomass and microbial communities of the plant rhizosphere were analyzed to examine the legacy effect of $H_2$ soil exposure. The legacy effect is defined here as a modification of the composition or activity of the microbial community during the activation stage that persists in the second stage of the experiment, without an external supply of $H_2$. The legacy effect includes stimulation of soil biological processes through plant-soil feedback caused by the plant growth-promotion effect of activated HOB. The activation of HOB did not support apparent growth of the HOB population and caused little impact on the composition of the soil microbial community. Elevated $H_2$ exposure reduced the below-ground biomass of *V. sativa* and no difference was noticed for the above-ground biomass. This manuscript was previously published as a preprint (bioRxiv 2025.02.12.637852; doi: https://doi.org/10.1101/2025.02.12.637852).

## MATERIAL AND METHODS

### Activation of HOB in soil microcosms

Sandy loam soil (80 g organic matter kg$^{-1}$, $C/N = 10$ and pH $= 6$) was collected in an experimental field trial as described by *Agoussar et al. (2021)*. Soil was homogenized and sieved (two mm mesh size). The water content was adjusted at 30% water holding capacity, and a defined amount (300 g) was transferred into 500 mL (nominal) Wheaton borosilicate bottles, with water content composing approximately half of the total volume. Bottles were sealed with a gas tight rubber septum cap. Two different $H_2$ treatments were applied: elevated $H_2$ concentration (e$H_2$) and atmospheric $H_2$ concentration (control). Soil exposure to e$H_2$ was achieved by injecting a defined volume of certified 99.999% compressed $H_2$ (Praxair Distribution Inc., PA, USA) in bottles to reach 5,000 ppmv in the static headspace. Injections were conducted every 2–3 days after 30 min of equilibration of the headspace with the ambient atmosphere. For the control treatments, bottles were left open for 30 min to equilibrate the headspace with atmospheric $H_2$ (approximatively 0.5 ppmv). Static headspace was replaced every 2–3 days. Microcosms were incubated for two months in dark conditions at room temperature to activate HOB (*De la Porte et al., 2024*; *Piché-Choquette et al., 2016*). Each experimental unit was represented by 20 repetitions (2 treatments × 20 repetitions = 40 microcosms). Soil subsamples (0.25 g) were collected after the incubation for DNA extraction, and the balance was transferred into pots for vetch cultivation.

### Hydrogen oxidation rate

After seven incubation days, $H_2$ oxidation rate was monitored sporadically. Measurements were performed on a random selection of three soil microcosms representative of each $H_2$ exposure treatment. A gas chromatographic assay was utilized to measure the first-order oxidation rate under 5,000 ppmv $H_2$ concentration (*De la Porte et al., 2024*). A defined volume of a certified gas mixture containing 100% $H_2$ (Distribution Praxair Canada, Saint-Laurent, Quebec, Canada) was injected into the static headspace of soil microcosms. First-order oxidation rates were computed by integrating $H_2$ time series recorded over 1 h (e$H_2$ treatment) or 2 h (control treatment). $H_2$ was measured with a gas chromatograph (Agilent Technologies, Santa Clara, California, USA) equipped with a thermal conductivity

detector. Calibrations were conducted using a certified $H_2$ gas mixture (10,000 ppmv $H_2$, balance air). Linear regression was integrated from three dilutions of the certified gas mixture: 0.1% (1,000 ppmv), 0.5% (5,000 ppmv), and 1% (10,000 ppmv) $H_2$.

## Vicia sativa pot experiment

Soil microcosms representative of each $H_2$ exposure were selected after the $H_2$ activation stage (10 repetitions × 2 treatments = 20 pots in total). Soil samples (approximately 300 g) were mixed with vermiculite (5:1, soil:vermiculite) and transferred to 1 L pots. *V. sativa* L. seeds (EBENA BRAND-FF 2023-0261, Meuneries Mondou) were surface disinfected and allowed to germinate in petri dishes containing 1% water-agar solution at 25 °C, 50% with a photoperiod of 16h/8 h light/darkness, respectively. Seedlings were transferred into the pots and randomly placed in a growth room under full spectrum LED lights for two months. Pots were regularly watered with tap water. The biomass of *V. sativa* was harvested to determine the fresh and dry weight of aerial and roots biomass. The rhizosphere soil was collected for subsequent DNA extraction, namely ''legacy-stage''.

## PCR amplicon sequencing and quality control of the bacterial 16S rRNA gene

Total DNA was extracted from soil microcosms after the activation stage and from rhizosphere soil after plant harvest with the PowerLyzer PowerSoil kit (Qiagen), following the instructions from the manufacturer. PCR amplicon sequencing of the bacterial 16S rRNA gene was performed with the primers 515F and 806R, targeting the V4 hypervariable region of the 16S rRNA gene (*Caporaso et al., 2011*). The procedure for library preparation was followed as described by *Saavedra-Lavoie et al. (2020)*. A few DNA samples were excluded during the library preparation process due to low DNA quality and quantity, leaving a total of 33 samples for the activation stage and 20 samples for the legacy stage. The sequencing and quality control were performed at the *Centre d'expertise et de service Génome Québec* (Montreal, Québec, Canada) with the Illumina MiSeq PE-250 platform. Downstream analyses of raw sequences were performed on RStudio software using Cutadapt for primer removal and the DADA2 pipeline for quality control, an amplicon sequence variant (ASV) table, and taxonomic assignation (*Callahan et al., 2016*; *Martin, 2011*). The taxonomic assignation was based on the SILVA v128 database (*Quast et al., 2012*). ASVs representing less than 0.005% relative abundance were not considered for downstream analysis. Species richness, Shannon diversity, and Simpson diversity were computed with iNEXT v3.0.1 (*Chao et al., 2014*; *Hsieh, Ma & Chao, 2016*). Raw sequence reads were deposited in the Sequence Read Archive of the National Center for Biotechnology Information under BioProject PRJNA1200744.

## Quantification of *hhyL* gene abundance

The abundance of the *hhyL* gene, encoding for the large subunit of [NiFe]-hydrogenases group 5/1 h, was determined in soil samples collected during the activation stage. The droplet digital PCR (ddPCR) assay described by *Baril et al. (2022)* was utilized. Randomly selected DNA from each treatment (control and e$H_2$) were used for the ddPCR assays. Negative controls were tested using DNA-free sterile water. The threshold was set manually

to include rain in the positive fraction (considering only those with up to 12% error), and samples with more than 10,000 droplets were selected for further analyses. Copy number concentrations were converted to copy per gram of dry soil.

## Statistical analyses

Statistical analyses were performed using the software Rstudio v4.2.3 (*Team, 2013*). Graphics were generated with ggplot2 v3.5.2 (*Wickham, 2016*). ASVs aggregated at the genus level were subjected to ANCOM-BC analyses to assess the response of bacteria to the activation phase using microbiome v1.20 and ANCOM-BC v2.0.3 (*Lahti & Shetty, 2018*; *Lin & Peddada, 2020*). Comparison of biomass dry weight (DW) between treatments and alpha diversity of microbial communities was examined with one-way ANOVA using de stats package v4.2. A *t*-test was computed to compare the absolute abundance of *hhyL* genes between the two $H_2$ exposure treatments. Raw datasets utilized for statistical analyses are provided as supplementary material (Tables S1–S8).

## RESULTS

### HOB activation stage

The $H_2$ oxidation rate measured in the control treatment was relatively stable, at $29 \pm 3$ nmol $g_{(soil-dw)}^{-1}$ $h^{-1}$ on average, without significant temporal pattern (Fig. 1). $H_2$ oxidation rate of the control soil decreased in weeks 6 and 7 prior to increasing again at weeks 9 and 10. Repetition of the experiment would be required to verify if the transient loss of activity was caused by technical issues or biological processes. No variations in soil water content or incubation conditions were noticed. Activation of HOB in microcosms exposed to elevated $H_2$ concentration was proved to be efficient with the $H_2$ oxidation rate varying from $47 \pm 8$ to $80 \pm 11$ nmol $g_{(soil-dw)}^{-1}$ $h^{-1}$ after 30 days of exposure to $63 \pm 5$ nmol $g_{(soil-dw)}^{-1}$ $h^{-1}$ (Fig. 1). The plateau of the $H_2$ oxidation rate indicates the energy potential of $H_2$ and nutrient content in soil were not sufficient to support proliferation of HOB. This conclusion was supported by indistinguishable *hhyL* gene copy numbers among control, with $2.2 \pm 1.7 \times 10^6$ copies per g, and e$H_2$ treatments, with $2.9 \pm 2.2 \times 10^6$ copies per g (*t*-test, *p*-value = 0.56; Table S5).

### Soil microbial communities

Most bacterial 16S rRNA gene sequences were affiliated with the phyla Actinobacteria (28%), Proteobacteria (27%), Thaumarchaeota (16%), and Acidobacteria (10%). The impact of $H_2$ exposure during the activation stage on soil bacterial communities was negligible, as variations of alpha diversity were not significantly influenced by the e$H_2$ treatment (Tables S2 and S3). Aggregation of the ASV at the genus level led to the identification of three genera displaying contrasting differential abundances between the two treatments. A higher log$_2$ fold-change (LFC) was observed in the e$H_2$ treatment compared to the control, and two groups were identified as being favored by $H_2$ exposure, namely *Aeromicrobium* spp. (LFC = 1.02, *p*-value = 0.002) from the Actinobacteria phylum and *Ramlibacter* spp. (LFC = 0.74, *p*-value = 0.00005) from the Proteobacteria phylum (Fig. 2). A single group of ASVs aggregated to the genus *Pseudoduganella* spp. appeared

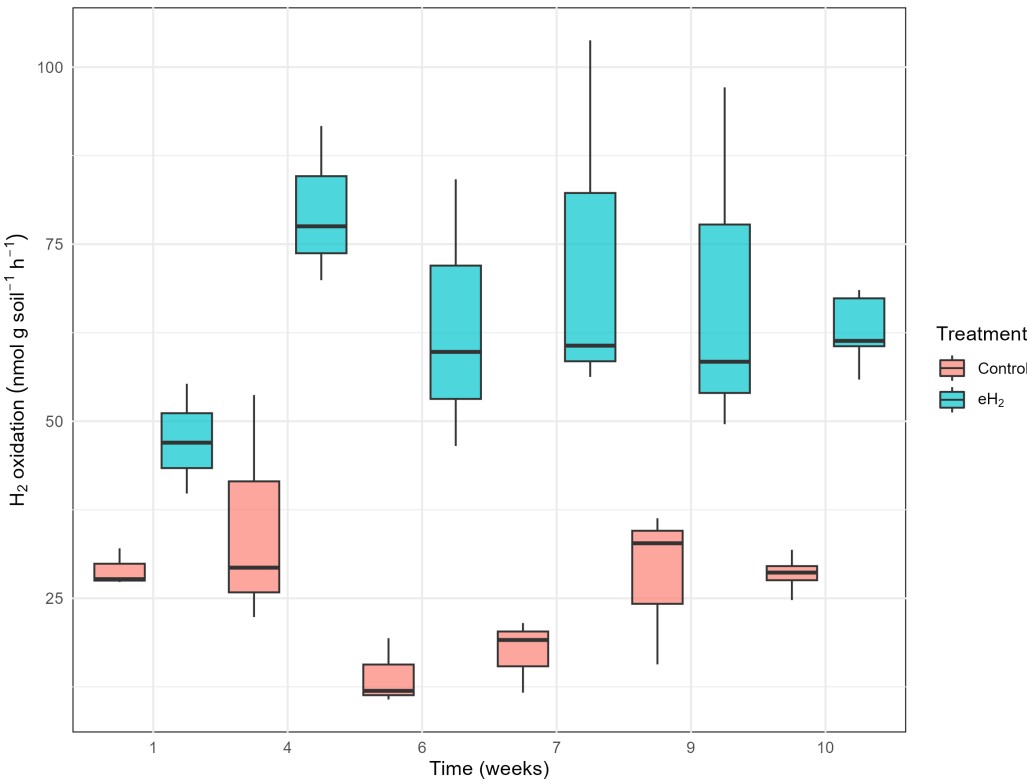

**Figure 1** **$H_2$ oxidation rates (nmol $g_{soil}^{-1}$ $h^{-1}$) in soil microcosms exposed to elevated (eH$_2$) or atmospheric (Control) $H_2$ concentration.** Each measurement routine was conducted on three randomly selected replicates during the activation stage of the experiments.

more favored by the control treatment than by the eH$_2$ treatment, but the difference was not significant (LFC $= -1.25$, $p$-value $= 0.08$). No legacy effect on the eH$_2$ treatment was observed on soil microbial communities in the rhizosphere of vetch as $H_2$ exposure neither influenced the alpha diversity nor the distribution of ASVs aggregated at the genus level (Tables S1 and S3).

### The legacy of HOB activation of vetch

The eH$_2$ treatment did not influence the aerial biomass of *V. sativa* (Fig. 3A), whereas it did reduce root biomass ($p$-value$= 0.034$; Fig. 3B). The root/shoot ratio, usually used as an observation of the plant response to the environment, was reduced in eH$_2$ exposure treatment ($p$-value $= 0.028$; Fig. 3C).

## DISCUSSION

The production of $H_2$ in $N_2$-fixing nodules comprising rhizobia not possessing [NiFe]-hydrogenase creates steep $H_2$ concentration gradients in the rhizosphere of legume plants. These concentration gradients expand at long distances, up to several centimeters away from nodules. In addition to redefining the spatial boundaries of the rhizosphere effect, these gradients are expected to play a role in modifying the microbial community structure

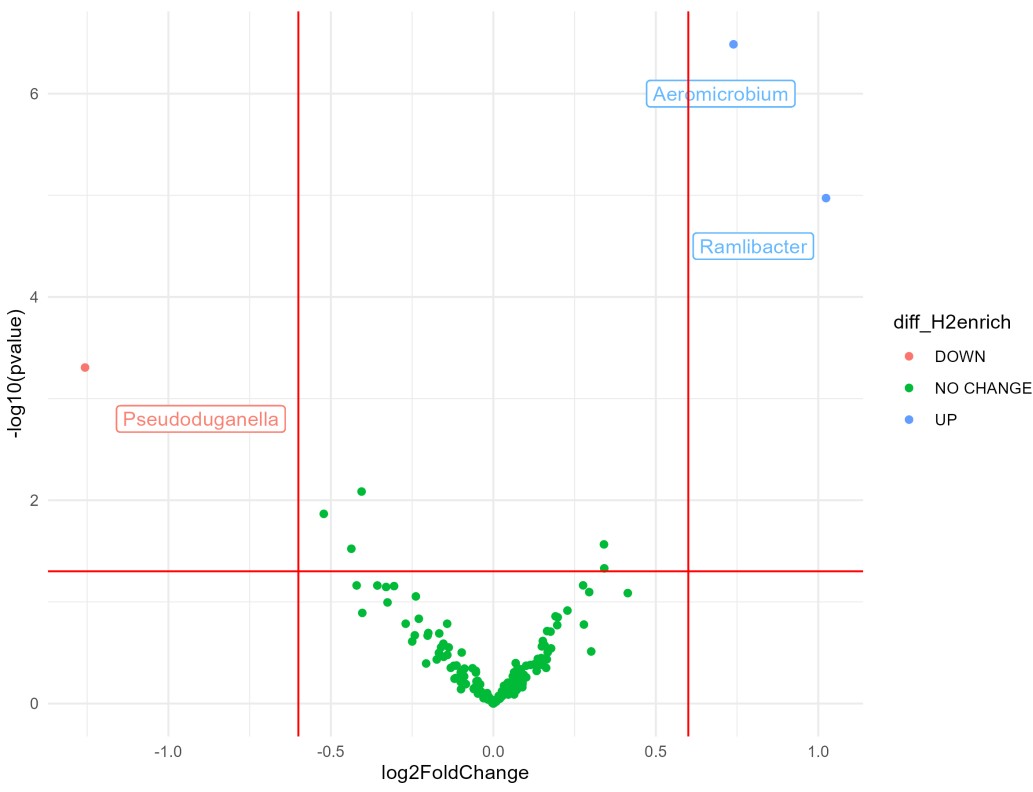

**Figure 2** **Volcano plot representing the ANCOM-BC analysis ($n = 10$) for the activation stage of microbial communities.** $Log_2$ fold change (eH$_2$/Control) of ASV aggregated at the genus level in response to H$_2$ enrichment at the activation stage of the experiments. Blue dots represent genera favored by eH$_2$ soil exposure (top-right quadrant), and the red dot represents the single genus favored in the Control treatment, meaning not favoured in the eH$_2$ treatment (top-left quadrant). Green dots show non-responsive bacterial genera to H$_2$ exposure (low-mid quadrant).

and function (*De La Porte et al., 2020*). This notion was first supported by the enrichment of HOB activity in soil surrounding N$_2$-fixing nodules in pioneering investigations performed in pot experiments (*La Favre & Focht, 1983*). Subsequent efforts were invested in artificial soil microcosms exposed to elevated H$_2$ concentrations. H$_2$ exposure enhanced net CO$_2$ fixation in soil, which is due to the facultative chemolithoautotrophic life strategy of numerous HOB in soil (*Stein et al., 2005*). The application of terminal restriction fragment length polymorphism (T-RFLP) led to the conclusion that only a few members of the microbial community were favored by H$_2$ exposure (*Osborne, Peoples & Janssen, 2010*). This observation was contrasted in subsequent investigations by applying high-throughput PCR amplicon and metagenomic sequencing techniques coupled with loose differential abundance analyses, suggesting H$_2$ exposure induced significant alteration of microbial communities, mainly in the rare biosphere (*Khdhiri et al., 2017*; *Piché-Choquette et al., 2016*). Revisiting these experiments with more stringent statistical tools tailored to the sparse microbiome data in the current study and in a recent work (*De la Porte et al., 2024*) confirmed conclusions supported by earlier T-RFLP techniques that H$_2$ exerts negligible impact on the soil microbial community. The exact mechanisms responsible for variations

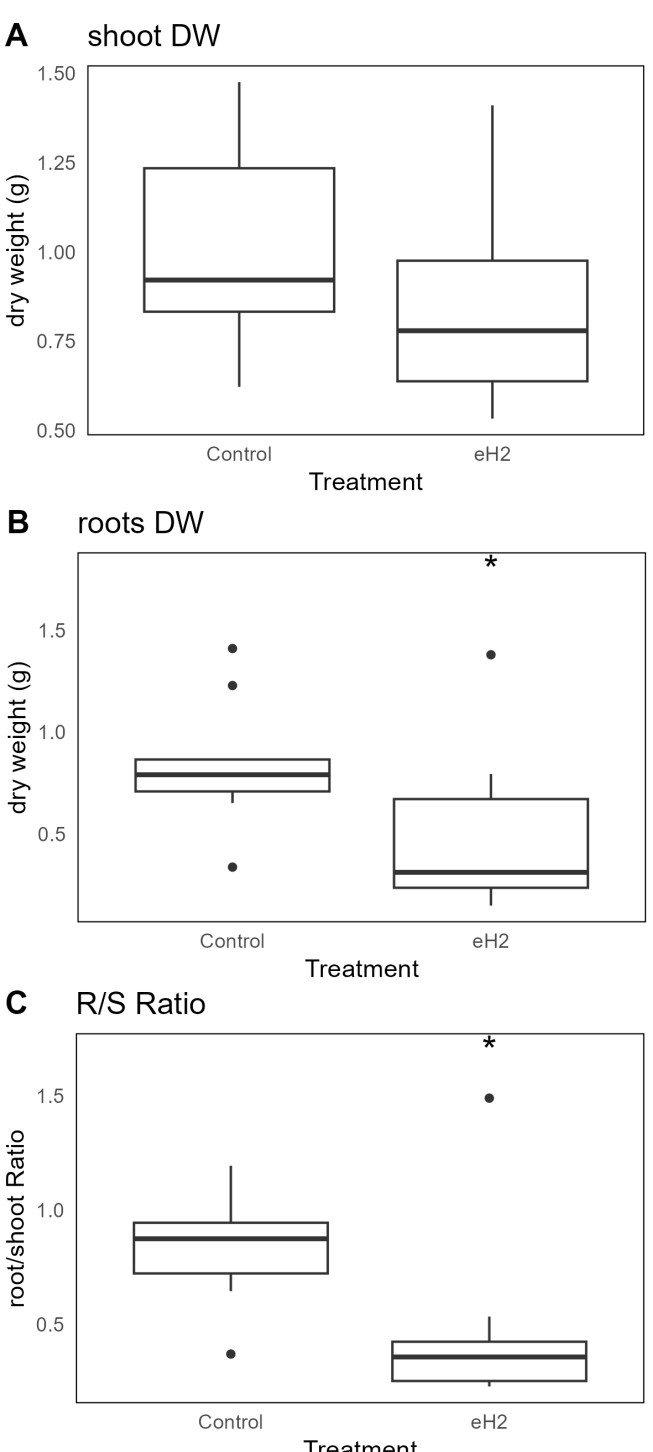

**Figure 3** **Differences in dry weight (g) of *V. sativa* after harvest in soil conditioned under elevated (eH₂) or atmospheric (Control) H₂ concentrations.** The dry weight (g) of (A) aerial, (B) root, and (C) root/shoot ratio is represented ($n = 10$).

in biogeochemical processes upon soil exposure to $H_2$ are therefore decoupled from HOB proliferation and compositional change of microbial communities.

Only two ASV groups, *Aeromicrobium* and *Ramlibacter,* were positively influenced by the $eH_2$ treatment during the activation stage (Fig. 2). *Pumphrey, Ranchou-Peyruse & Spain (2011)* reported the enrichment of labelled DNA from a phylotype affiliated with *Aeromicrobium ginsengisoli* from hairy vetch rhizospheric soil incubated with 100 ppm of $H_2$. There is, therefore, evidence for chemolithoautotrophic metabolism in *Aeromicrobium* spp., but demonstration of the activity in isolates has not been reported. In addition to the negligible alteration of microbial communities by $H_2$, there is also no evidence supporting the proliferation of HOB encoding for group 1H/5 [NiFe]-hydrogenase in soil exposed to $eH_2$ as quantification of *hhyL* was indistinguishable between treatments (Table S5). The enrichment of HOB encoding [NiFe]-hydrogenase from other groups cannot be ruled out as they were not detected in the quantitative PCR assay. For instance, HOB encoding for [NiFe]-hydrogenase encompassing group 2a (*Bay et al., 2021*; *Cordero et al., 2019*; *Islam et al., 2020*), group 1f (Hyo type; *Myers & King, 2016*), group 1l (Hyl type; *Ortiz et al., 2021*), and group 1d (6C type; *Pandelia, Lubitz & Nitschke, 2012*) were not detected by *hhyL* PCR assays and are potential contributors of the $H_2$ oxidation activity measured in the current study. The stabilization of the $H_2$ oxidation activity in soil exposed to $eH_2$ was suggested in a previous microcosms experiment where the response of high-affinity $H_2$ oxidation activity plateaued after a one-week $H_2$ exposure period (*Piché-Choquette et al., 2016*). Extended incubation under a headspace enriched with $H_2$ was insufficient to induce a secondary enrichment after the enrichment plateau (Fig. 1). The level of $CO_2$ in the static headspace of soil microcosms during the activation stage as well as organic carbon and other nutrients were likely insufficient to support the proliferation of HOB.

Another aspect of $H_2$ exposure in agroecosystems is the fertilization effect of this gas. This was first reported in pot experiments where soil substrate was fumigated with $H_2$ prior to planting (*Dong et al., 2003*). The plant biomass increased by 15–48% in fumigated soil compared to controls. The fertilization effect was attributed to HOB displaying the plant growth-promotion effect (*Maimaiti et al., 2007*), such as 1-aminocyclopropane-1-carboxylate (ACC) deaminase activity reducing the accumulation of ethylene, which acts as a stress phytohormone (*Yang & Hoffman, 1984*). This is indirect evidence suggesting that recruitment of symbiotic $N_2$-fixing rhizobia releasing $H_2$ ($Hup^-$) is an evolutionary advantage for legume plants. Rhizobia possessing [NiFe]-hydrogenase ($Hup^+$) to recycle $H_2$ produced by nitrogenase exist, but they appeared less represented by their $Hup^-$ counterparts in nature (*Annan et al., 2012*). Consideration of the fertilization effect of $H_2$ and the selection of Hup- symbioses in nature led to the suggestion that $H_2$ could be a missing link in clarifying why there are multiple benefits of crop rotation that have not been explained by nitrogen fertilization alone (*Golding & Dong, 2010*).

The benefits of $H_2$ on plants are likely multifactorial. The fertilization effect was not observed for wheat grown in soil subjected to fumigation by $H_2$, CO, or $CH_4$ (*De la Porte et al., 2024*). The aerial biomass of vetch in the present study was not enhanced in soil conditioned with $eH_2$ exposure treatment (Fig. 3). The reduction in root biomass, as well as in root/shoot ratio, is thought to be related to the plant growth-promotion activity of

HOB, including ACC deaminase reducing root proliferation for nutrient prospection. If this mechanism holds true, it was realized in the absence of evident HOB proliferation. These results must be considered in trials integrating irrigation treatments with $H_2$-saturated water, enhancing stress resistance in plants. The antioxidant potential of $H_2$, microbial activation of immune system in plants, and HOB are potential contributors to plant benefits (*Wang et al., 2024*). A more holistic analysis, examining both plant and microorganism interactions in soil exposed to different $H_2$ levels or in contrasting Hup+/Hup- symbioses, is recommended to disentangle the contributions of microorganisms and plant metabolism in explaining the benefit of $H_2$.

## CONCLUSIONS

In conclusion, soil exposure to $eH_2$ led to an activation of HOB without supporting their growth. $H_2$ was therefore sufficient to support the persistence energy requirement of HOB, probably by rewiring their energy metabolism toward an inorganic energy source and a repression of carbon metabolism. Examination of the potential synergetic effect of $H_2$ with nutrients will decipher the limiting factors for HOB proliferation and activation in soil.

### Funding

This work was supported by a Discovery grant (RGPIN-2024-06451) from the Natural Sciences and Engineering Research Council of Canada (NSERC) to Philippe Constant. The funders had no role in study design, data collection and analysis, decision to publish, or preparation of the manuscript.

### Grant Disclosures

The following grant information was disclosed by the authors:
Natural Sciences and Engineering Research Council of Canada (NSERC): RGPIN-2024-06451.

### Competing Interests

The authors declare there are no competing interests.

### Author Contributions

- Diana Dip conceived and designed the experiments, performed the experiments, analyzed the data, prepared figures and/or tables, authored or reviewed drafts of the article, and approved the final draft.
- Philippe Constant conceived and designed the experiments, authored or reviewed drafts of the article, and approved the final draft.

### Data Availability

The raw sequence reads are available in the Sequence Read Archive, National Center for Biotechnology Information: BioProject PRJNA1200744.

The raw data is available in the Supplemental Files.

## Supplemental Information

Supplemental information for this article can be found online at http://dx.doi.org/10.7717/peerj.20019#supplemental-information.

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
