# Peer review of "The potential biofertilization effect of H2 is accompanied by a modest impact on the composition of microbial communities in the rhizosphere of common vetch"

_PeerJ, doi:10.7717/peerj.20019_

## Round 0.1 · original submission · Minor Revisions

We have received two highly detailed and very constructive reviews. Reviewer 1 found the manuscript interesting but called for major revisions in Methods. Reviewer 2 noted the need to fix grammatical errors and provide references. They also called for more description of the metadata. I suggest a providing an Rmarkdown file of the data analysis pipeline. I provide some examples of suggested revisions below.
Since I did not see a clear need for collecting more data or the need for a reanalysis of the presented results, I decided the manuscript needs minor revisions.

Specific Comments

Abstract appears incomplete. State methods used to generate Results and provide a brief Discussion.
Line 36. Delete lines 36 – 37, start paragraph on topic (“N2-fixing…”).
Line 42. Here and throughout, avoid the passive voice. Revise to “At least 40 phyla of HOB exploit…”
Line 45. Avoid phrases like “it was estimated that.” Revise to “.. carbon and ##- 12 % of soil bacteria … to fix carbon….”
Line 50. Revise to “Soil exposure to 250 nmol H2 g(soil)-1 h-1 enriches CO2 fixation activity from HOB”
Line 52. What is the source of the statement that H2 enrichment enriches for CFB. Also, why is the method (in situ hybridization).
Line 54. Write succinctly. Replace “was also proven efficient to promote” with “promotes”
Line 58 – 71. This section is wordy.
Line 72 -73. Revise to: “This work was conducted to determine high concentrations of H2 in soils activates and increases the population of HOB.”
Line 74- 75. Revise to follow the same sentence structure of line 72 – 73 and join them together (…and if activation of HOB enhances the growth…”
Line 82. End Introduction with a clear statement of the key result of the work.
Line 125. Move “Raw reads…PRJNA1200744” to the end of the paragraph.
Line 127. Revise to “Sequencing and quality control..”
Line 147. Cite source of ggplot (Wickham 2016)
Line 157-160. Do not repeat results. Delete “The activation…modified atmosphere.”
Line 160. Do not start sentence by directing reader to figure. State the result “H2 oxidiation…(Fig. 1)
Line 164. Delete “at the end of the incubation period.”
Line 167. Write actively “…hhyL gene copy numbers…supports…” Also, move this to Discussion.
Line 185. As above, do not start paragraphs in Results with restatement of methods. Start by stating the result.
Figure 1. Box or violin plots are more appropriate than bar charts. Remove grid lines. Subscript 2 in H2. Superscript -1 in y-axis label.
Figure 2. Remove gray gridlines and explain red lines.
Figure 3. As Figure 1, remove gray gridlines and subscript 2 in H2.

·

Basic reporting

The manuscript tests how molecular hydrogen impacts community structure and subsequent plant performance by incubating soil with and without 0.5% hydrogen, sequencing 16S rRNA, and growing vetch. Analysis of 16S rRNA gene assays show some increased populations though they remain largely unchanged (Fig 2). Plant performance decreased with elevated hydrogen concentrations (Fig 3). Hydrogen usage rates are presented (Fig 1). There was no observed legacy effect and community abundance did not increase under eH2 conditions (hhyL abundance). This suggests elevated hydrogen does not always increase plant performance. The manuscript focused on the HOB community shift and potential legacy effect while also presenting data that suggests the H2 biofertilization effect is not universal.

There is no geochemical or enzymatic/nutrient data to understand how hydrogen would impact nutrient availability abiotically – the paper mentions this limitation in the conclusion and shifts it focus on the HOB community to place proper emphasis on their results. I think this makes the manuscript more convincing. The manuscript is limited to community analysis and hydrogen usage with plant performance and thus doesn’t draw any conclusions on the mechanisms for the lower observed plant performance; rather it just uses the performance results to show that H2 biofertilization is not universal. The manuscript has an interesting goal, relevant cited literature, and sufficient data to support its conclusions – it was a good read. I have some various comments to further improve the readability below:

Clear language is used throughout (though I have minor comments to improve readability). There is a great amount of literature cited and the reader will sufficiently understand the problem and aims of the manuscript. Raw data is accessible but needs more explanation for names in the excel sheet. Some parts of the figures need refinement.

Figure 1: Using a bar plot for this data seems suboptimal but the authors may have good reasons. Would a box plot (like Fig 3) over time not be better? This could help show the distribution differences. Or a scatter/series plot over time with error bars with some sort of loess line. If you really prefer the current figure, it is not that big of an issue for readability, it just doesn’t play to the strengths of bar plots. Regardless, the y-axis must be labelled correctly (hydrogen oxidation rate).

Figure 2: n = ?. The key needs to be redone. The legend name is very uninviting and the treatment conditions need to be labelled accurately (“no” should be “no change”). The red lines which separate significant change should be described in the figure caption. This plot is very convincing, I am a fan.

Figure 3: These box plots are very effective as well. Avoid abbreviations when possible (DW = dry weight) to increase readability.

Are these plots not exported at 300 DPI? Perhaps it is just the website integration, but the quality is lower than it should be for a manuscript on my screen. Ensure the quality is exported at 300 DPI or some higher quality.

Experimental design

Hydrogen oxidation rates would benefit from written methods more than just citing de la Porte, 2024. Considering this would be an open source manuscript and the cited manuscript is not, this limits the availability of methods (I couldn’t even access the manuscript, I had to borrow it from the shared library system on behalf of another university). Considering how tricky hydrogen is to measure, the methods need to be explained in detail. The rest of the methods were quite clear and the reader should be able to replicate them. The use of so many replicates is wonderful and the experimental design is rigorous.

Validity of the findings

The main conclusions are:
1. HOB abundance does not increase with elevated hydrogen.
a. This is due to CO2, nutrients, and organic carbon limitation
2. There is little population change
3. Plant performance is decreased
a. Therefore H2 biofertilization is not fully understood and was not observed here
4. No legacy effect observed

I searched for the evidence for point 1 in the supplemental data , but it is unclear what samples MC or MH are or what experiment DQ means. I recommend adding some more explanation to the supplemental data to explain variable names.

Points 2 and 3 are very clear to me. Good job.

The conclusions on no observed legacy effect are not apparent to me. Either the manuscript didn’t sufficiently prepare the reader to know what the legacy effect is and what evidence would support it, or the evidence is presented and the conclusions are not drawn in an apparent way for the reader due to missing reasoning in the discussion. This conclusion may very well be supported in the manuscript, but it was not easily understood by me.

The discussion was well researched and explained the data well, it is just that the points above need refined.

Additional comments

- Defining “legacy effect” would help the reader.
- When discussing data, it would be very helpful to the reader if you cite the figure which supports that data or include data in parentheses. This is especially true when discussing no hhyL change, as there is no figure to help the reader. A simple repeat of the results (2.2 vs 2.9 units) would further support the point while discussing the implications. This also helps the reader distinguish what conclusions are from your data, which are from literature, and which are reasoned through from data but not supported directly.
- The manuscript swaps between the terms "control and eH2" vs "control and H2" at various points in the manuscript and figures. Standardizing this to eH2 would be better.
- The conclusion introduces new references and reads more like a discussion than a conclusion. Consider moving this to the discussion or adding emphasis on results directly from this manuscript.
Miscellaneous readability issues, referenced by line:
Lines 42-43: reads odd, recommend rewording
Line 54: define PCB or omit that detail if not necessary
Lines 72-82: is this really all fit for the introduction? It feels like a lot of detail, though I agree with ending the introduction with a brief explanation of what is to come in the rest of the manuscript.
Line 99: Missing/extra parenthesis, “Soil subsamples (0.25 g)”
Line 177: define LFC
Line 253-255: Reword.

I would be happy to read another version of this manuscript after the hydrogen oxidation methods are added, the supplemental data is explained further, and the legacy effect results are explained further.

Gage R. Coon

·

Basic reporting

Overall, the manuscript by Dip and Constant, provided an interesting insight into the potential legacy effect of hydrogen oxidising bacteria on common vetch. The introduction of the manuscript provides good context for the scope of research carried out, and mostly references relevant past studies. However, there are numerous sentences in the introduction and discussion for which references are missing that reference previously generated data (by either this group or others), namely L44-45; 47, 551-53, 66-68, 196-197.

It was interesting to see that the elevated hydrogen condition seemed to have little effect on the alpha diversity of the microcosms under investigation, as well as the hydrogen consumption rates. Data described in the deposition statement has been deposited correctly, noted in the manuscript and accessible from the NCBI Sequence Read Archive BioProject PRJNA1200744. Raw data has been provided for the microcosm experiments in the form of an Excel file. There are some small issues with grammar throughout the manuscript, particularly in plural subject-verb agreements (e.g. in the Figure 1 legend), and some other typographic errors e.g. missing bracket in L99, wrong word used (L53 should be impact, L208 should be negligible) or missing word (L104 should have the word 'gas' before chromatographic).

Whilst data was provided for the abundance of the hhyL gene and the alpha diversity effects in the manuscript results section, the actual data presented in the Supplementary were not appropriately referred to in text i.e. the manuscript was missing statements as to where the data was presented in L169, L182. Figure resolution could be improved as all provided figures were slightly blurry.

Thank you for providing the raw data, however your supplemental files need more descriptive metadata identifiers to be useful to future readers. In particular, in the contents page of the Supplementary Excel file, it would be good to include full-length table legends with full descriptors of what the sample names are referring to or alternatively provide an additional file of Supplementary table legends, as it is currently very difficult to understand which sample relates to what treatment.

Experimental design

The manuscript contains original primary research within the scope of the journal, and the experiments performed have been mostly well replicated. Though there are some remaining questions surrounding why only some of the DNA was randomly selected for use in various downstream analyses rather than using the whole set of extracted DNA (e.g. L139), which would increase the reliability of the results obtained. The results of the research correlate well with the original research question and their determined research gap.

As the hhyL gene abundance methodology is specifically written out, it would have been good to have seen this data as a standalone graphical figure rather than just as one short text line and the data in tabular form in the Supplementary Material. Additionally, can the authors please comment on why only the hhyL hydrogenase subtype was focused on? There are numerous other uptake hydrogenases that have previously been identified in soil, yet these were not mentioned or tested for. Whilst the concentrations of hydrogen chosen for the microcosm were stated to select for high-affinity hydrogen oxidisers, it is possible that low-affinity hydrogen oxidisers could have also been enriched here.

There appears to be less samples for the legacy stage alpha diversity (20) calculations than the activation stage (33). Can the authors please comment on why that is the case? Additionally, it is not very clear from the methods section when the "activation" and "legacy" stages are in terms of soil collection/extraction - are these before and after incubation with the elevated H2, or before and after the plant trial?

From the methods section, it is not very clear where the soil used for the microcosms came from. The authors reference a 2021 study for the method of collection, but they fail to state where their soil was collected from, when it was collected, how it was stored prior to use and whether any physicochemical analysis was done on the soil to determine baseline nutrient levels. This is particularly relevant for the pot experiment, as if the soils were low in key nutrients e.g. nitrogen, this could be responsible for the negative plant effects observed. As per the methods currently (L110-118), there was no added fertiliser to the pots. Additionally, in L165-166, the authors state that the H2 oxidation plateau could be due to insufficient nutrient content in the soil, but they do not provide data to back this up.

Validity of the findings

All underlying data have been provided, and have been appropriately statistically analysed. The robustness and replication level of the data are mostly good, except for the aforementioned case where only randomly selected DNA samples were used for the ddPCR quantification.

The conclusion stated that soil exposure to eH2 was not sufficient to support the growth of HOB as previously hypothesised. However, as the authors only targeted on subtype of hydrogenase (Group 1h/5, hhyL) in their ddPCR, this statement only holds true for this group of HOB, not HOB in general as there are far more subtypes of uptake hydrogenases than assayed in this study. As such, I suggest that the authors clearly state that this is the case earlier in their manuscript, or tone down this statement.

Additional comments

L174: wrong supplementary table referred to. Table S1 is the ASV count not the alpha diversity indices. Additionally, the table legends provided for the supplementary material do not have enough information i.e. what does MC, MH, VC, VH and the subsequent numbers refer to?

Supplementary material: It would be better to also present the alpha diversity indices and the hhyL copy number data in graphical form rather than just tabular as it is hard to determine the significance or insignificance of these results when grouped by treatment in tabular form. Column A is missing the title from Table S2, S3, S4.

If vials were incubated with ambient air, this would not typically be classified as 'subatmospheric' levels of hydrogen. Please reword throughout the manuscript to reflect the true availability of the hydrogen in these controls (ambient hydrogen).

Can the authors please common on why they think that the H2 oxidation rate of the control soil drastically decreases in weeks 6 and 7 prior to increasing again at weeks 9 and 10 in Figure 1?

---

## Round 0.2 · Minor Revisions

The consensus is that the manuscript is close to acceptable. Please address the minor revisions suggested by reviewer 2.

Regards,

Michael

**Language Note:** The review process has identified that the English language must be improved. PeerJ can provide language editing services - please contact us at [email protected] for pricing (be sure to provide your manuscript number and title). Alternatively, you should make your own arrangements to improve the language quality and provide details in your response letter. – PeerJ Staff

·

Basic reporting

The manuscript has been clearly improved through the peer review process and I recommend its publication. The figures and accessibility of supplemental data have been improved.

Experimental design

The improved hydrogen method section makes it clear how data was collected and analyzed.

Validity of the findings

no comment

·

Basic reporting

Overall, I am mostly satisfied with the changes to the manuscript that have been enacted since the previous review. However, there are some aspects of the manuscript that still do not adequately address the comments provided in the previous review.

L39-40 (Abstract – results): for the sentence beginning “The root biomass…”, the phrase after the comma is currently quite unclear. Please amend. It is either a grammatical error (reduces) or there are missing words in the sentence.

There are still numerous grammatical errors present within the manuscript. Please carefully go through it to eliminate instances of plural subject-verb disagreements, missing words and spelling errors. Some examples are as follows, but this is not an exhaustive list and extends throughout the entire mansucript: L59 (Incubation of soil samples under “a” modified atmosphere), L61 (soil microbial communities), L67 (led “to” variable impacts), L69 (influence?), L77 (missing comma before the “and”), L81 (concentrations), L83 (modified atmospheres), L85 (either add “the” before “microbial community” or make it “communities”), L86 (an external supply), L89 (a minor impact on the composition of soil microbial communities), L110 (into pots), L137 (missing space between words), L279 (random bracket and extra full stop).

L87: There are numerous errors in this sentence, and as it currently reads, the meaning of the sentence is very unclear. Please rephrase it. I presume it should be “stimulation” rather than “simulation”? and it should be “processes” not “processed”. “Plant-soil feedback expressed” is a very confusing phrase.

Experimental design

L35-240: The existence of other high affinity HOB outside of just the hhyL hydrogenase subtype needs to be made clearer. Whilst the manuscript has now included one sentence on the lack of enrichment of other high affinity HOB hydrogenase subtypes (L239), this was only framed as a limitation of the ddPCR assay rather than a true discussion about other types of HOBs. A targeted ddPCR assay would not detect other subtypes as only one primer set was used, so it would be impossible to detect other subtypes. More qualifying statements to this effect are needed to appropriately address the limitation of the study in not considering other types of HOB, such as in the results subsection of the abstract (L35)– “the abundance of hhyl HOB was indistinguishable…” and in the discussion section.

L88: The physicochemical reporting of the soils used in the microcosm are similarly not addressed in enough detail. Whilst a carbon-nitrogen ratio was now provided (line 88), there was still no comment on the baseline level of nitrogen in the pot nor discussion about how low levels of nutrients could be contributing to the negative effects observed as there was no additional fertiliser added to soils that have been stored at 4°C for over 3 years.

L96: sieving and homogenising are different steps in the processing of soil. Please rephrase.

Validity of the findings

More qualifying statements are needed to appropriately address the limitation of the study in not considering other types of HOB.

Additional comments

Table S4: line A5, what is the taxon here? There is just a number (45597)

---

## Round 0.3 · accepted · Accept

The reviewer's comments were addressed and I am happy with the current version. The manuscript is accepted.